# Evaluation of AIML + HDR—A Course to Enhance Data Science Workforce Capacity for Hispanic Biomedical Researchers

**DOI:** 10.3390/ijerph20032726

**Published:** 2023-02-03

**Authors:** Frances Heredia-Negron, Natalie Alamo-Rodriguez, Lenamari Oyola-Velazquez, Brenda Nieves, Kelvin Carrasquillo, Harry Hochheiser, Brian Fristensky, Istoni Daluz-Santana, Emma Fernandez-Repollet, Abiel Roche-Lima

**Affiliations:** 1RCMI-CCRHD Program, Medical Sciences Campus, University of Puerto Rico, San Juan 00934, Puerto Rico; 2Department of Public Health, Medical Sciences Campus, University of Puerto Rico, San Juan 00934, Puerto Rico; 3Department of Biomedical Informatics, University of Pittsburgh, Pittsburgh, PA 15260, USA; 4Department of Plant Science, University of Manitoba, Winnipeg, MB R3T 2N2, Canada; 5Department of Biostatistics and Epidemiology, Medical Sciences Campus, University of Puerto Rico, San Juan 00934, Puerto Rico

**Keywords:** data science, artificial intelligence, machine learning, health disparities, hispanic biomedical research

## Abstract

Artificial intelligence (AI) and machine learning (ML) facilitate the creation of revolutionary medical techniques. Unfortunately, biases in current AI and ML approaches are perpetuating minority health inequity. One of the strategies to solve this problem is training a diverse workforce. For this reason, we created the course “Artificial Intelligence and Machine Learning applied to Health Disparities Research (AIML + HDR)” which applied general Data Science (DS) approaches to health disparities research with an emphasis on Hispanic populations. Some technical topics covered included the Jupyter Notebook Framework, coding with R and Python to manipulate data, and ML libraries to create predictive models. Some health disparities topics covered included Electronic Health Records, Social Determinants of Health, and Bias in Data. As a result, the course was taught to 34 selected Hispanic participants and evaluated by a survey on a Likert scale (0–4). The surveys showed high satisfaction (more than 80% of participants agreed) regarding the course organization, activities, and covered topics. The students strongly agreed that the activities were relevant to the course and promoted their learning (3.71 ± 0.21). The students strongly agreed that the course was helpful for their professional development (3.76 ± 0.18). The open question was quantitatively analyzed and showed that seventy-five percent of the comments received from the participants confirmed their great satisfaction.

## 1. Introduction

Data science (DS) is an interdisciplinary field that uses scientific methods, processes, and algorithms to extract knowledge and insights from structured and unstructured data across various application domains [1]. Artificial intelligence (AI) is an area of computer science that considers building intelligent machines capable of performing tasks that typically require human intelligence [2]. One branch of AI is machine learning (ML), which incorporates algorithms to receive data inputs and predict outputs within preselected ranges and patterns [2]. Currently, there are some biases in the existing DS approaches for minority healthcare solutions. For example, Obermeyer et al., 2019 [3] demonstrated that a commercial ML algorithm developed to guide healthcare decisions falsely concluded that African-American patients were less in need of medical care than equally sick Caucasian patients. This occurred because the black population represented in the dataset received fewer medical treatments based on their socio-economic status [3]. While data science approaches have the potential to help predict diagnosis and treatments, they can also exacerbate existing health disparities and inequalities if the data is not adequately driven [4]. For example, Vyas et al., 2020 [5] found some race-adjusted algorithms that “perpetuate or even amplify race-based health inequities” through the interpretation of racial inequalities as immutable biological facts [5].

One of the strategies for resolving or preventing health disparities (HD) in health-related data science applications is by employing more diversity in the teams of data scientists that develop these algorithms. Having diverse opinions, ideas, and sentiments can help teams strive for better and more robust solutions that consider each point of view while also being more creative. Additionally, diversity leads to better problem-solving, expands the talent pool, and is essential for long-term economic growth [6]. Unfortunately, minority populations are disproportionately represented in data science research, workforce participation, and available data. Although Hispanics are one of the largest minority groups in the United States (18%), there is not enough collected and open data for this group [7,8]. In addition, Hispanics represent only 8% of the total US enrollment in data science or related fields, while most participants (50%) are of Caucasian ethnicity [9].

The future workforce in science and medicine is an essential subject to focus on. To mitigate health disparities, educating and inspiring biomedical experts from various backgrounds is crucial. Lunn et al. establish the importance of culturally competent care, meaning that it is essential that the workforce with experience in the community acquire knowledge, attitudes, and skills to address health disparities [10]. There are multiple approaches to teaching about health disparities, but the topic of data science is usually not considered. For example, there is a course based on helping medical students become aware of their biases toward racial and ethnic minorities that is taught as a 5-day elective. It employs didactic lectures, group discussions, expeditions to local clinics and hospitals, and poster session workshops [11]. Another approach to addressing health disparities is to integrate the topic into the undergraduate curriculum to create a biomedical workforce that knows about the issue and wants to help throughout their careers [12]. A different example of a course related to health disparities included the intersectionality of sexual orientation, gender identity, and race and ethnicity. In this course, the students received a short lecture and videos of patients of color describing their experiences [13]. None of the courses included in their curriculum address the topic of data science.

The lack of diversity in data science education needs to be addressed to mitigate health disparities [14]. The current approach has been focused on developing DS courses for minority students, but the courses do not teach HD explicitly. Most commonly, the courses focus on machine learning and AI, specifically natural language processing, classification, tree-based algorithms, machine learning, and AI [15]. There is one Data Science for Health Disparities Research course that is not taught explicitly to minority students [16]. Minorities need to be specifically targeted and trained to identify and mitigate the disparities that disproportionately affect their communities [17].

As DS and ML become an integral part of the healthcare process, the need to have a diverse workforce trained in both DS and HDR will be crucial. There is an urgency to strengthen and enhance the diversity of the workforce. In May 2021, the National Institute of Health—National Institute for Minority Health Disparities (NIH-NIMHD) released a Notice of Special Interest (NOSI) entitled “Administrative Supplements to Enhance Data Science Capacity at NIMHD-Funded Research Centers in Minority Institutions (RCMI)”. This grant became the funding source for the submitted proposal, “Implementation of the Artificial Intelligence and Machine Learning applied to Health Disparities Research (AIML + HDR) course with Focus on the Hispanic population”. This proposal aimed to create and teach a new course to enhance and build DS capacities for biomedical investigators and graduate students at the University of Puerto Rico (UPR), mainly at the Medical Science Campus (MSC). Participants were trained in DS, AI, and ML topics, such as Jupyter Hub, coding with Python, using ML libraries, creating predictive models, and other cutting-edge techniques to identify and address bias and thus mitigate Hispanic health disparities. A final survey was distributed to the participants as part of the course. This paper aims to describe and evaluate the results from the participants’ course assessments as well as their recommendations to enhance and build DS capacities in minority populations. 

## 2. Materials and Methods

### 2.1. Course Curriculum

The AIML + HDR course was designed and implemented in two parts (Table 1). The first part included theoretical concepts about artificial intelligence, machine learning, health disparities, and data sources (Units 1–4). The second part focused on developing the participant’s skills in R and Python programming languages (Units 6 and 7). Finally, Units 8 and 9 included the implementation of predictive machine learning models using Hispanic datasets based on the five phases of the DS project lifecycle [18]. 

New didactic materials were created in each unit, such as presentations, lectures, demonstrative videos, and reading materials [19,20,21,22,23,24,25,26,27,28,29,30]. The written course materials were prepared in English, but most of the videos were recorded in Spanish. The professors and staff were from the UPR-MSC. In addition, there were invited speakers from the University of Pittsburgh, USA, and the University of Manitoba, Canada. AIML + HDR was implemented as an asynchronous course in the RCM-Online TalentLMS platform [31]. The course was available for 15 weeks (April to July 2022), where participants had access to all the materials and were expected to review all the content and submit the corresponding evaluations.

### 2.2. Course Participants 

Ninety-six candidates from UPR completed the application form with an interest in participating in the course. As defined in the initially submitted proposal, the following priority criteria were considered to select the course participants: the candidate is (1) an early career stage investigator at University of Puerto Rico-Research Center of Minority Institution) program (UPR-RCMI), (2) an active investigator at UPR-RCMI, (3) technical staff UPR-RCMI, (4) an early-stage investigator at UPR-MSC, (5) an investigator at UPR-MSC, (6) a technical staff at UPR-MSC, (7) a graduate student, researcher, or staff conducting biomedical research or practice on the Hispanic population at UPR. Finally, 34 participants were chosen to take the course based on the above criteria and the course staff’s availability to assist and support the learning process. All participants were native Spanish speakers but fluent in English.

### 2.3. Course Assessment

At the end of the online AIML + HDR course, a survey was given to the participants to assess their satisfaction with the organization, platform, teaching resources, and recommendations to enhance and build DS capacities in minority populations. The assessment instrument was based on an existing survey [32] used by the UPR-RCMI to evaluate educational activities. The survey was modified to assess specific online course aspects of the covered units, language, and other elements related to the organization and content. A total of 17 participants completed the survey.

The survey (Table 2) was divided into five general questions that evaluated the following elements: (#1) structure and design of the course, (#2) carried out activities, (#3) other elements, (#4) covered DS topics, and (#5) general comments and suggestions. The total questionnaire (Table 2) included 20 premises with a four-point Likert scale [33]. The answers ranged from “Totally Disagree” or “Very Unsatisfied” (ordinal-scale value = 1), “Mostly Disagree” or “Mostly Unsatisfied” (ordinal-scale value = 2), “Mostly Agree” or “Mostly Satisfied” (ordinal-scale value = 3), to “Totally Agree” or “Very Satisfied” (ordinal-scale value = 4). In addition, one open-ended response question for comments and suggestions was included. 

### 2.4. Statistical Analysis

Descriptive statistics were primarily used to analyze participant information, quantitative survey data, and the open-ended question. The participant demographics and background information were totaled, and averages and percentages were obtained. Quantitative survey data (Questions #1–#4) was based on ordinal-scale values (ranging 1–4) to obtain the percentages, average, and standard deviation for each question. Cronbach’s alpha coefficient [34] was used to estimate the internal consistency of all the questions. 

An open-ended question (Question #5) was related to participants’ comments and suggestions. To quantitatively analyze this question, the comments were classified into the three categories that the students mentioned: (1) “general satisfaction with the course”, (2) “topics suggested for future courses”, and (3) “requests for improvements in the current course”. For each comment, there was a classification value by category. For example, for category (1) “general satisfaction with the course”, a comment was classified as 1 for high satisfaction, 0 for a neutral comment or comment that did not address satisfaction, and −1 for negative comments or comments that indicated dissatisfaction with the course. Then, the numbers by categories were counted and tabulated, and the mean and other descriptive statistics were computed. 

## 3. Results

### 3.1. Course Participants

The total course participants were 34 Hispanic members of the different campuses at UPR. Figure 1 describes the demographics and other information of the participants. Sixty-two percent were men, and thirty-eight percent were women. Seventy-one percent of the students belonged to the Medical Sciences Campus, 15% to the Mayagüez Campus, 12% to the Rio Piedras Campus, and 3% to the Bayamón Campus. There was a greater representation of graduate students (32%) and staff (29%—including professors, technicians, postdocs, or resident physicians). Early-stage investigators (18%), other interested personnel (8%), and undergraduates (3%) also benefited from the course. From the pool of participants, twenty-seven percent are working at the RCMI-Center for Collaborative Research in Health Disparities.

### 3.2. Course Assessment—Quantitive Survey Questions (Questions #1–4)

Ordinal-scale values from the quantitative questions (Questions #1–#4) were used to compute the internal consistency for all items, resulting in a Cronbach’s coefficient of 0.95. 

#### 3.2.1. Question #1—Structure and Design of the Course

Through the completion of a survey, participants evaluated the course. Question #1 results are described in Figure 2 and assess the structure and design of the course, indicating the participant’s level of agreement with the statements. The responses were recorded on a Likert scale where Totally Disagree = 1, Mostly Disagree = 2, Mostly Agree = 3, and Totally Agree = 4. In the first statement, “The course is easy to handle”, the results were 72% “mostly agree” and 38% “totally agree”, equaling a mean value of 3.29 ± 0.21. When asked about “The course is well organized” and “The readings and resources are adequate to learn about the topics of the course”, the results were 38% “Mostly agree” and 67% “totally agree”, resulting in a mean value of 3.71 ± 0.21 on both statements. For the last question, “The work plan is easy to understand and follow”, 6% of students “mostly disagree”, 38% “mostly agree”, and 61% “totally agree”, equaling a mean value of 3.59 ± 36.

#### 3.2.2. Question #2—Course Activities

Figure 3 shows the results for the second group of statements (Question #2) regarding the assessments of the course activities, such as tests and tasks. The first statement was “Course activities are varied”, where 58% of participants “mostly agree”, 37% “totally agree”, and 5% “mostly disagree”, equaling a mean value of 3.20 ± 0.21. For the statements “Activities are relevant to the course” and “Activities promote my learning”, the results were 68% “totally agree” and 32% “mostly agree”, equaling a mean value of 3.71 ± 0.21 on both statements. To the premise “Activities include clear and easy-to-understand instructions”, the participant answers were 53% “totally agree”, 37% “mostly agree”, and 11% “mostly disagree”, equaling a mean value of 3.41 ± 0.48.

#### 3.2.3. Question #3—Different Course Elements

For the statements in Question #3, the participants were asked about their general level of satisfaction with different course elements (Figure 4). The responses were recorded on a Likert scale: Very Unsatisfied = 1, Mostly Unsatisfied = 2, Mostly Satisfied = 3, and Very Satisfied = 4. For the premise “The learning experience in the course”, the results were 50% for both “mostly satisfied” and “totally satisfied”, equaling a mean value of 3.53 ± 0.25. For the statements “The usefulness of the course for my professional development” and “Course resources”, 70% of the participants were “totally satisfied” and 30% were “mostly satisfied”, equaling a mean value of 3.76 ± 0.28 for both statements. Concerning the “Activities to evaluate my learning”, 10% of the participants were “mostly unsatisfied”, 60% were “totally satisfied”, and 40% were “mostly satisfied”, equaling a mean value of 3.53 ± 0.37.

#### 3.2.4. Question #4—Course Materials 

Figure 5 shows the results concerning Question #4, where participants were asked about their satisfaction with the course materials, specifically for each unit. Overall, the highest satisfaction was with “Unit 3: Health Disparities Data Types and Hispanic Datasets”, with 80% feeling “very satisfied” and 20% feeling “mostly satisfied”. The other units that stood out were “Unit 1: Introduction to Artificial Intelligence and Machine Learning”, “Unit 2: Applying Artificial Intelligence and Machine Learning in Health Disparities”, and “Unit 5: Introduction to Machine Learning Projects”, with 30% and 70% of the participants feeling “mostly satisfied” and “very satisfied”, respectively. For “Unit 4: Introduction to Bioinformatics”, “Unit 8: Understanding and Data Preprocessing”, and “Unit 9: Model Planning, Model Building, and Results”, 40% of the participants were “mostly satisfied” and 60% were “very satisfied”. “Unit 6: Introduction to Programming (Python and R)” and “Unit 7: Advanced Programming (Python)” had the lowest satisfaction, with 10% of the participants feeling “mostly unsatisfied”.

Regarding satisfaction with the covered topics in the course, we also computed the average and standard deviation for the ordinal-scale values of Question #4 (Table 3). As can be seen, the topic relating to health disparities was consistently rated highest (highest average and lowest standard deviation). Additionally, theoretical or conceptual topics (Units 1–5) were rated higher than practical or programming units (Units 6–9). 

### 3.3. Question #5—Open-Ended (Comments and Suggestions) 

Question #5 asked participants to share their comments and/or suggestions about the course and how it could be improved. Figure 6 shows the results from quantitatively evaluating these comments and suggestions. Seventy-five percent (75%) of the comments that were received from the participants confirmed their great satisfaction. Some examples of positive comments were: “*Excellent course! It was very organized, and materials were clear and concise. Definitely, I will take another course from this team.*”; “*The course was well organized. Overall, it was a great activity.”;* and “*I really enjoyed this course!*” Eight percent (8%) of the comments were negative, mostly indicating the course duration was not sufficient to complete the exercises on time. 

Regarding suggested topics for future courses, two participants made a total of three suggestions. These suggestions included (1) reinforcement learning and deep learning methods applied to health, (2) cloud technologies, and (3) advanced Python programming. Regarding the requests for improvements, 33% of the students made one suggestion. These suggestions included fixing typos, requesting the course be taught synchronously, and adding more practical and challenging exercises. 

## 4. Discussion

This introductory course focused on strengthening the knowledge of participants from UPR in DS, AI, and ML and their applications to health disparities research. Some technical areas included the Jupyter Notebook Framework, coding with R and Python to manipulate data, and ML libraries to create predictive models. The goal of this course was to mitigate the lack of a knowledgeable biomedical workforce by applying these topics to health disparities research with a focus on Hispanic health. Some of the topics covered were: Electronic Health Records and how this data or lack of data can affect treatment choices; Social Determinants of Health and how they affect health outcomes; how inequalities can be identified and addressed; and how to identify bias in data and address it. The course participants also acquired skills to generate and use Hispanic datasets, making them Findable, Accessible, Interoperable, Reusable (FAIR) and AI/ML-ready [35]. Furthermore, they learned how to apply ML methods to create relevant and personalized predictive models for medical treatments and diagnosis for minority health, specifically Hispanic populations.

The UPR was chosen as the institution to teach this course because it is a minority-serving institution where 98% of the staff and students are Hispanic [36]. Although 96 people were interested in taking the course, only 34 were selected to participate. Most participants (71%) were from the Medical Science Campus because there were pre-defined criteria to select the students when the proposal was submitted to NIH-NIMHD. These criteria included being part of the “Research Center of Minority Institutions” (RCMI) program and conducting biomedical research or practice on the Hispanic population. 

The course was assessed with a survey that included four questions with 20 premises rated on a four-point Likert scale and an open-ended question for comments and suggestions. Figure 7 summarizes the most relevant answers that show the course’s impact. Regarding the premises “*The course is well organized,*” “*Activities promote my learning,*” and “*The usefulness of the course for my professional development*”, we obtained that 100% of the participants “*Mostly agreed*” or “*Totally agreed.*” This result shows high satisfaction with the course. In relation to “*The work plan is easy to follow,*” 61% of the participants “*totally agreed*” and 33% “*mostly agreed.*” However, there were still 6% that “*mostly disagreed.*” Considering these last answers and the fact that some participants did not complete the course, we should revise the work plan to make it easier for students to follow.

Question #4 of the survey was about the individual units of the course. The health disparities units were the highest rated, thus supporting the premise that a focus on Hispanic disparity-related issues is needed. This is consistent with another report where the Health Disparities course received the highest rating of any course at that medical school, indicating that medical school students are very receptive to the concept of health inequities and ways to address them [11]. Theoretical or conceptual topics were rated higher than practical or programming units, suggesting that the participants felt more confident learning the coding concepts in Python, which makes sense considering their biological backgrounds. We should modify future courses to make the programming and practical units more attractive by incorporating other examples and activities. 

All the participant recommendations and suggestions from Question #5 were valuable. Negative comments were related to the course duration and lack of time to complete all the activities. This type of negative comment is typical for online or asynchronous courses, where other research [37,38] has reported similar comments. Overall, the participants were very grateful for the experience. The team will continue upgrading courses to improve the experience for future course editions.

Limitations: As the survey was incorporated at the end of the course, only the participants who completed more than 95% of the course activities (a total of 17 participants) could access the survey. Although it is representative of the course assessment, considering this is the first course taken by Hispanic biomedical researchers to be trained on applying DS, AI, and ML to Hispanic health disparities. Another limitation of the course is the gender imbalance (62% males and 38% females), which contrasts with the usual 50% gender distribution in a medical school [39]. This imbalance can be attributed to the fact that computer science-based courses and related fields are likely to attract more males [40]. It will be considered for future course editions.

## 5. Conclusions

This course, AIML + HDR, impacted the development of the biomedical workforce at the UPR-MSC, a Hispanic Serving Institution, by developing DS, AI, and ML skills on Hispanic datasets, such as making and using the FAIR dataset and creating predictive models for diagnosis and treatments. In general, the results of the assessment were positive. Most of the participants provided excellent feedback about the organization and structure of the course and its impact on their professional development. This course has enhanced the data science skills of participants from the UPR biomedical research community, allowing them to be more successful in obtaining competitive extramural support, particularly for research that disproportionately impacts the health of Hispanic populations. In addition, early-stage investigators, post-doctoral fellows, and graduate students that were part of this course have incorporated this knowledge into their development phases to improve future research on Hispanic minority health and health disparities. So far, two graduate students have included ML approaches in their dissertation research because of their participation in this course. 

This project is expected to impact the general Hispanic community in Puerto Rico and the United States. The focus of the AIML + HDR course on advancing the science of minority health and health disparities will also have a significant impact on improving the health of the Puerto Rican population. Participants are trained in the importance of health disparities research, factors that contribute to health disparities, and applications of DS, AI, and ML to mitigate and achieve health equity.

## Figures and Tables

**Figure 1 ijerph-20-02726-f001:**
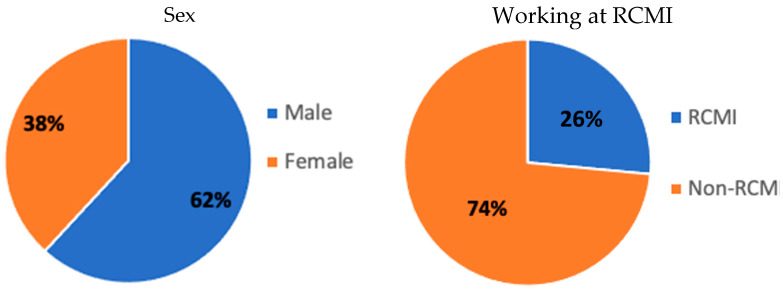
Course participants’ demographics and other information.

**Figure 2 ijerph-20-02726-f002:**
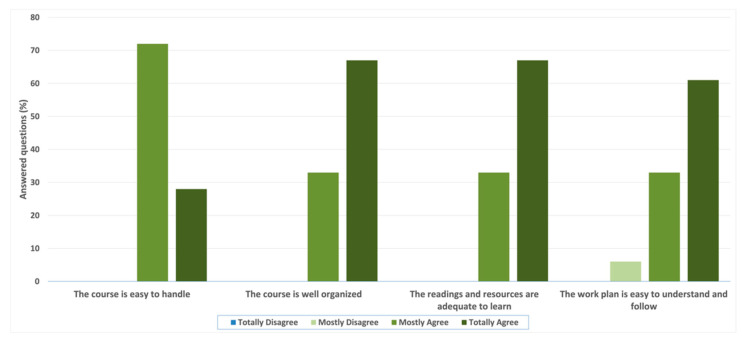
Participant’s answers to Question #1—“Indicate your level of agreement with the following statements about the structure and design of the course”.

**Figure 3 ijerph-20-02726-f003:**
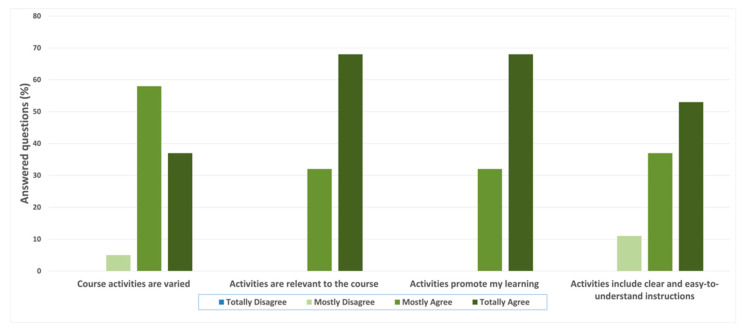
Participant’s answers to Question #2—“Indicate your level of agreement with the following statements about the activities (tests, tasks) of the course”.

**Figure 4 ijerph-20-02726-f004:**
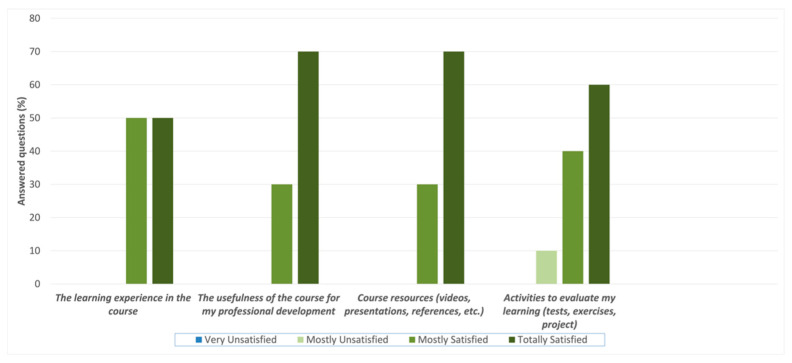
Participant’s answers to Question #3—“Indicate your general level of satisfaction with the following course elements”.

**Figure 5 ijerph-20-02726-f005:**
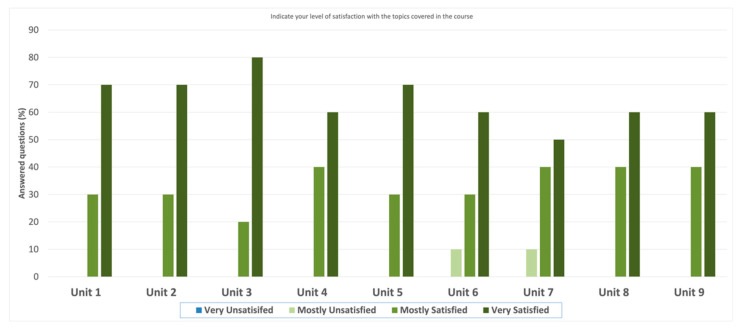
Participant’s answers to Question #4—“Indicate your level of satisfaction with the topics covered in the course”.

**Figure 6 ijerph-20-02726-f006:**
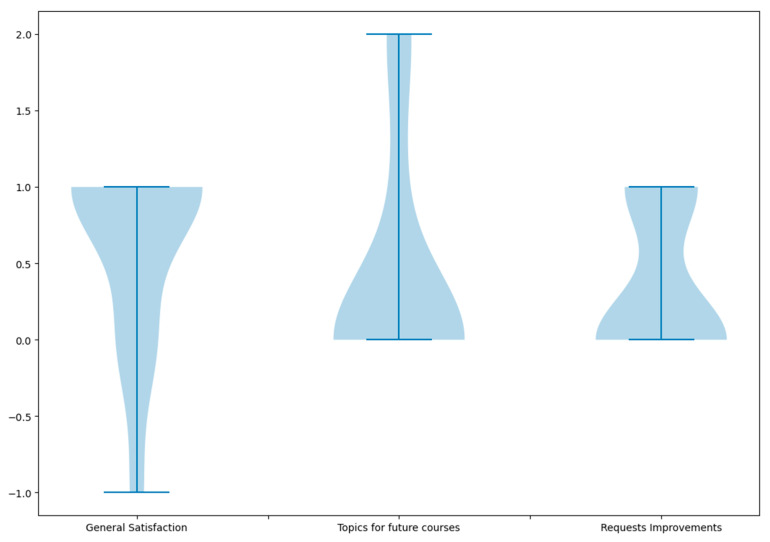
Quantitative interpretation of participant’s answers to open-ended Question #5—“Comments and/or suggestions”.

**Figure 7 ijerph-20-02726-f007:**
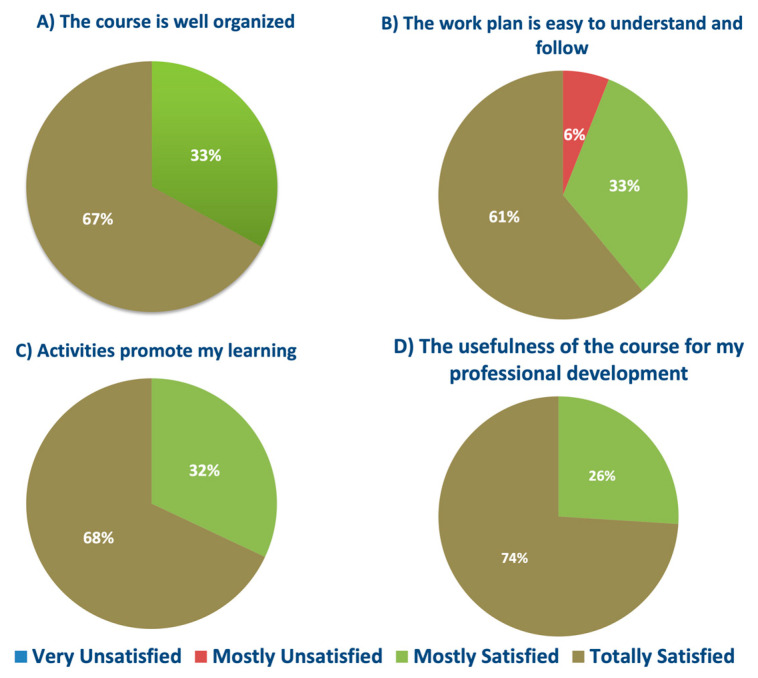
Overview of the most relevant questions: (**A**) “Q#1.2—The course is well organized”, (**B**) ” Q1.4—The work plan is easy to understand and follow”, (**C**) “Q#2.3—Activities promote my learning”, and (**D**) “Q#3.2—The usefulness of the course for my professional development”.

**Table 1 ijerph-20-02726-t001:** Course content outline.

Part I—Introductions and Data Sources
Unit 1—Introduction to Artificial Intelligence (AI) and Machine Learning (ML) Unit 2—Applying AI/ML Tools to Mitigate Health Disparities Unit 3—Health Disparities Data Types and Hispanic Datasets Unit 4—Introduction to Bioinformatics Data Unit 5—Introduction to Machine Learning Projects
Part II—Creating Predictive Models for Health Disparities Research
Unit 6—Introduction to Programming (Python and R) Unit 7—Advanced Programming—Python Unit 8—Understanding and Data Pre-processing—Python Unit 9—Model Planning, Model Building, and Communication Results—Python

**Table 2 ijerph-20-02726-t002:** Course participants questionnaire.

Question Number	Survey Question	Type of Question
#1	Indicate your level of agreement with the following statements about the structure and design of the course:	[Multiple select] Choices: Totally Disagree Mostly Disagree Mostly Agree Totally Agree
	Q1.1. The course is easy to handle Q1.2. The course is well organized Q1.3. The readings and resources are adequate to learn about the topics of the course Q1.4. The work plan is easy to understand and follow	
#2	Indicate your level of agreement with the following statements about the activities (tests, tasks) of the course:	[Multiple select] Choices: Totally Disagree Mostly Disagree Mostly Agree Totally Agree
	Q2.1. Course activities are varied Q2.2. Activities are relevant to the course Q2.3. Activities promote my learning Q2.4. Activities include clear and easy-to-understand instructions	
#3	Indicate your general level of satisfaction with the following course elements:	[Multiple select] Choices: Very Unsatisfied Mostly Unsatisfied Mostly Satisfied Very Satisfied
	Q3.1. The learning experience in the course Q3.2. The usefulness of the course for my professional development Q3.3. Course resources (videos, presentations, references, etc.) Q3.4. Activities to evaluate my learning (tests, exercises, project)	
#4	Indicate your level of satisfaction with the topics covered in the course:	[Multiple select] Choices: Very Unsatisfied Mostly Unsatisfied Mostly Satisfied Very Satisfied
	Q4.1. Unit 1: Introduction to Artificial Intelligence (AI) and Machine Learning (ML) Q4.2. Unit 2: Applying AI/ML Tools to Mitigate Health Disparities Q4.3. Unit 3: Health Disparities Data Types and Hispanic Datasets Q4.4. Unit 4: Introduction to Bioinformatics Data Q4.5. Unit 5: Introduction to Machine Learning Projects Q4.6. Unit 6: Introduction to Programming (Python and R) Q4.7. Unit 7: Advanced Programming Q4.8. Unit 8: ML for Hispanic Datasets—Understanding and Data Wrangling Q4.9. Unit 9: ML for Hispanic Datasets—Model Planning, Model Building, and Communication Results	
#5	Q5.1. Comments and suggestions	[Other—Text]

**Table 3 ijerph-20-02726-t003:** Evaluation for course topics (Question #4) using the average (Avg.) and standard deviation (Std.) of the ordinal-scale values.

Question #4	Avg.	Std.
1. Unit 1: Introduction to Artificial Intelligence (AI) and Machine Learning (ML)	3.76	0.18
2. Unit 2: Applying AI/ML Tools to Mitigate Health Disparities	3.71	0.21
3. Unit 3: Health Disparities Data Types and Hispanic Datasets	3.82	0.15
4. Unit 4: Introduction to Bioinformatics Data	3.65	0.23
5. Unit 5: Introduction to Machine Learning Projects	3.71	0.21
6. Unit 6: Introduction to Programming (Python and R)	3.59	0.36
7. Unit 7: Advanced Programming	3.53	0.37
8. Unit 8: ML for Hispanic Datasets—Understanding and Data Wrangling	3.59	0.24
9. Unit 9: ML for Hispanic Datasets—Model Planning, Model Building, and Results	3.59	0.24

## Data Availability

All data generated from this research are presented in this manuscript.

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
