# Peer review of "Evaluation of AIML + HDR—A Course to Enhance Data Science Workforce Capacity for Hispanic Biomedical Researchers"

_ijerph, 2023, doi:10.3390/ijerph20032726_

Round 1

Reviewer 1 Report

Thank you for providing an opportunity to review the manuscript. This is an interesting study that provides an asynchronous course "Artificial Intelligence and Machine 17 Learning applied to Health Disparities Research (AIML+HDR)" that applied general Data Science 18 (DS) approaches to health disparities research with emphasis on Hispanic populations. The text is relatively well written; however, it needs major improvement:

1)     The introduction is incomplete. Authors should provide the information on the course of AI and ML about the method, pros, and cons in previous studies among the minority in other countries or continent or other populations.

2)     Authors should present and cite the articles that related to the AL and ML education and training.

3)     Please shows how to calculate the sample size and what is the reference using in calculating this?

4)     Out of 34 participants, 17 completed the survey. My concerns are only 50% of participants completed the survey which might not be enough to representative the minority population even in this setting.

5)     Authors should present more clearer in the methodology such as how to sum the score of each element and interpret them instead of presenting each item. In addition, authors should classify the total score into categories such as Low or high of level of agreement.

6)     Please shows the reliability (alpha) of the questionnaire.

7)     The authors did not declare the ethical issue in this study.

8)     In the results, the imbalance of the proportion of UPR campus should be explained.

9)     The colour of the bar chart should be blended from the same colour.

10)  The result from open end question should be analysed by qualitative technique.

11)  In the discussion, I cannot evaluate until adjust all the comment in the methodology and result.

12)  Please add more references and discussion to compare with previous studies (now just only one study has been referred) in the discussion part.

Reviewer 2 Report

While going through the paper titled “Artificial Intelligence and Machine Learning Applied to Health Disparities Research (aiml+hdr): a Course to Enhance Work- force Capacity for Hispanic Biomedical Researchers” I found it interesting, but it is more pretending a course pack rather than a complete manuscript. The topic and contents are very nicely discussed but still I have some comments which may improve the quality of your paper.

The abstract should be upgraded, less information about the context and more quantitative of information about the study. Methodology and findings should be provided in detail.

In introduction section, there is a need to connect the paragraph with each other so that there should be a sense of continuity of arguments. Furthermore, the arguments given in the introductions section need to be strengthen through citation of the recent research articles and significance of the proposed novelty (aim1+hdr). In this section, please highlight the novelty by adding few more lines. The objectives should be more clear so that the reader can understand the direction of your research. For improving your work, please support your discussion through advance figures (rather than simple bar and pie charts) and explain through verbal arguments.

The literature review part is very week, it should compare the study and develop the research gap ultimately. The method you have used for estimation purpose is quite appropriate but it need the justification of using it in your work. The results need to be supported through the previous research work justification and logic while interpretations needs more clarity. Quality of the figures and tables are quite appropriate. It should be precise and consistent with the objectives of the study.

Overall, the paper seems to be quite satisfactory, but you have not used artificial intelligence and machine learning techniques in the manuscript. So, you should remove AI/ML from the title of the following article. Your efforts are expressing development of course instead of developing new AI/ML techniques. Even your article has not shown any novelty that will be a value addition in the literature.

Round 2

Reviewer 1 Report

Thank you for providing an opportunity to review the manuscript again. The authors make significant changes and edit regarding all the requested changes and suggestions. This is an interesting study that provides an asynchronous course "Artificial Intelligence and Machine Learning applied to Health Disparities Research (AIML+HDR)" that applied general Data Science (DS) approaches to health disparities research with emphasis on Hispanic populations. The text is relatively well written; however, it needs minor improvement:

1)     Figure 7 is not explained and referenced from the paragraph.

2)     Please move table 3 to sub-section 3.2.4 Question #4 and move the paragraph “In terms of satisfaction with the covered topics in the course….” Into this sub-section as well. But please remain the discussion of these issues “This is consistent with another report…” in discussion part.

3)     Authors should explain what is the definition of “advanced more than 95%” in page 12?  

4)     Please add more the limitation about imbalance of population in this study such as male (62%), etc.
